# The Cell Culture Medium Affects Growth, Phenotype Expression and the Response to Selenium Cytotoxicity in A549 and HepG2 Cells

**DOI:** 10.3390/antiox8050130

**Published:** 2019-05-14

**Authors:** Lisa Arodin Selenius, Marita Wallenberg Lundgren, Rim Jawad, Olof Danielsson, Mikael Björnstedt

**Affiliations:** 1Division of Pathology, Department of Laboratory Medicine, Karolinska Institutet, Karolinska University Hospital Huddinge, S-141 86 Stockholm, Sweden; lisa.arodin@ki.se (L.A.S.); marita.wallenberg-lundgren@ki.se (M.W.L.); rim.jawad@ki.se (R.J.); olof.danielsson@ki.se (O.D.); 2Division of Clinical Microbiology, Department of Laboratory Medicine, Karolinska Institutet, Karolinska University Hospital Huddinge, S-141 86 Stockholm, Sweden; 3Division of Clinical Research Center, Department of Laboratory Medicine, Karolinska Institutet, Karolinska University Hospital Huddinge, S-141 86 Stockholm, Sweden

**Keywords:** cell culture, culture media, selenium, proliferation, cytotoxicity, differentiation

## Abstract

Selenium compounds influence cell growth and are highly interesting candidate compounds for cancer chemotherapy. Over decades an extensive number of publications have reported highly efficient growth inhibitory effects with a number of suggested mechanisms f especially for redox-active selenium compounds. However, the studies are difficult to compare due to a high degree of variations in half-maximal inhibitor concentration (IC_50_) dependent on cultivation conditions and methods to assess cell viability. Among other factors, the variability in culture conditions may affect the experimental outcome. To address this, we have compared the maintenance effects of four commonly used cell culture media on two cell lines, A549 and HepG2, evaluated by the toxic response to selenite and seleno-methylselenocysteine, cell growth and redox homeostasis. We found that the composition of the cell culture media greatly affected cell growth and sensitivity to selenium cytotoxicity. We also provided evidence for change of phenotype in A549 cells when maintained under different culture conditions, demonstrated by changes in cytokeratin 18 (CK18) and vimentin expression. In conclusion, our results have shown the importance of defining the cell culture medium used when comparing results from different studies.

## 1. Introduction

Redox-active selenium compounds have very interesting growth modulatory properties and are therefore highly relevant drug candidates for cancer therapy [1,2]. Dissimilarities in the culture conditions for a single cell line, make comparisons of the response to selenium cytotoxicity from different studies difficult as evident from vast variations in reported half maximal inhibitory concentration (IC_50_) of a specific compound in the literature (Table 1).

One explanation for the differences in cellular response may be related to interactions between the specific compounds of interest with the various bioactive molecules that vary between different culture media. While the suppliers of commercially available cell lines provide recommendations for optimal growth conditions, the selection and use of growth medium varies widely (Table 2).

Commercially available cell culture media contains buffers, inorganic salts, glucose, amino acids, vitamins and numerous bioactive compounds but the levels of each may differ substantially. For instance, the concentration of glucose in the media varies between 5.5 mM and 25 mM (Minimum Essential Medium (MEM) vs. Dulbecco’s Modified Eagle’s Medium (DMEM)), and the presence of ferric nitrate in DMEM and copper, iron and zinc sulphate in F12 medium, might interact with specific metal dependent enzymes of importance in cellular reactions.

The cell culture medium RPMI 1640 was developed for the culture of human leukemia cells and included glutathione and high concentrations of vitamins as well as biotin, vitamin B12, and 4-aaminobenzoic acid (PABA), which were not present in MEM or DMEM. In contrast, Ham’s F12 was developed to be used without serum for single cell cultures of hamster ovarian cells, and therefore comprised of a different formula. The general recommendations are to use MEM for cultivation of adherent cells and RPMI 1640 for suspension cells/hematopoietic cells [24,25].

The tumor microenvironment is of outermost importance in toxicity studies. Kim et al. demonstrated that cell confluence, concentration of fetal bovine serum (FBS) and different cell culture media modulated the gene expression in a breast cancer cell line [26]. Furthermore, the microenvironment has been shown to be crucial for selenium uptake and cytotoxicity and is thus a highly relevant factor to consider for toxicity evaluations of selenium compounds [27].

In this study, the two commercially available cell lines A549 (lung cancer), and HepG2 (hepatocellular carcinoma) were cultured in four commonly used culturing media (RPMI, F12, DMEM, and MEM). The aim of the present study was to investigate if these four cell culture media could influence the sensitivity to selenium cytotoxicity, proliferation, morphology, expression of markers for epithelial and mesenchymal differentiation, thiol levels and the expression of the selenoenzyme Thioredoxin reductase.

## 2. Materials and Methods

### 2.1. Chemicals and Reagents

Sodium selenite, selenomethylselenocysteine (MSC), Trizma^®^ base, Guanidine hydrochloride, 5,5-dithiobis(2-nitro-benzoic acid) (DTNB), bovine serum albumin (BSA), ethylenediaminetetraacetic acid disodium salt (EDTA), poly-ethylene-glucol (PEG), and H_2_SO_4_ were all purchased from Sigma-Aldrich (Darmstadt, Germany). RPMI 1640 medium (GlutaMAX™, HEPES), Ham’s F-12 nutrient mix (GlutaMAX™) DMEM (Dulbecco’s Modified Eagle Medium, high glucose, HEPES, GlutaMAX™), MEM (Minimum Essential Medium, HEPES, GlutaMAX™) and Fetal Bovine Serum (FBS) (South America origin) were all purchased from Gibco, Life Technologies (Paisley, United Kingdom). Horseradish peroxidase (HRP) conjugated streptavidin, and 3,3´,5,5´tetramethylbenzidine (TMB) substrate were purchased from Mabtech AB (Stockholm, Sweden).

### 2.2. Cell Culturing and Proliferation Rate

The cell lines A549 (ATCC) and HepG2 (ATCC) were cultured at 37 °C, under a humidified atmosphere of 5% CO_2_. All cells tested negative for mycoplasma and were subsequently grown in fresh medium (RPMI, F12, DMEM, or MEM) supplemented with 10% FBS. The same batch of FBS was used for all the experiments. Cells were cultivated for a minimum of two or maximum of seven passages in new media before experiments were initiated. Cells were counted and seeded at the same density during maintenance and before every experiment. To determine the toxicity of selenite and MSC in the different media, cells were seeded in 96-well plates and incubated for 24 h. Before the experiments were initiated, optimal cell density was investigated by seeding cells at different concentrations (cells/mL) and evaluated visually in light microscope at 24 and 48 h. Cell confluence at the treatment point should be near 70% for best treatment effect with selenium compounds, and to minimize the risk of over confluence at harvest time and analysis. The optimal seeding density in the experiments were: A549; 80,000 cells/mL and HepG2; 120,000 cells/mL (if not indicated otherwise). Cells were washed with phosphate-buffered saline (PBS) and treated with MSC or selenite in their new media. After 48 h, the cellular ATP content was measured using CellTiter Glo^®^ luminescent cell viability assay (Promega, Madison, WI, USA) following the manufacturer’s instruction. The luminescence was measured in a 96-well plate reader (BIOTEK FLx 800, Gen5 Data Analysis Software, BioTek, Winooski, VT, USA). Initially, toxicity experiments were performed using tetrazolium salt 4-[3-(4iodophenyl)-2-(4-nitrophenyl)-2H-5-tetrazolio]-1,3-benzene disulfonate, (WST-1) (Abcam’s Quick Cell Proliferation Assay, ab65473, Abcam, Cambridge, UK) following the manufacturer’s protocol, with 1 h of incubation time before analysis.

### 2.3. Viability Measurements

For determination of cell growth in the different media, cells were seeded at the same density at day zero (described above) and counted using trypan blue exclusion assay (Abcam, Cambridge, UK) in an automated cell counter (TC20^TM^ automated cell counter, BioRad, Stockholm, Sweden). After four days in culture, the cells were harvested, diluted in the same volume as when seeded and viable cells (considered as proliferating cells) were counted. Growth rate was determined as the fold change compared to number of cells seeded.

### 2.4. Intracellular and Extracellular Thiols

Cells were seeded and incubated for 48 h. At the day of the experiment, the cells were washed carefully with PBS and fresh medium was added, with a subsequent incubation of 5 h. For the determination of extracellular thiols, 1 mL of medium was collected from each flask and stored on ice until analysis. Cells were harvested using trypsin, washed in PBS and sonicated (3 × 5 s, with amplitude set at 40, Vibra-Cell, Sonics & Materials Inc. Danburg, CT, USA) in Tris-EDTA-buffer (TE buffer) kept on ice. Protein fractions were collected after centrifugation at 17.0× *g* for 30 min (MICROSTAR 17R, VWR, Leuven, Belgium). The thiol content was determined directly in the freshly prepared samples (as described in [27,28]). In brief, the reaction was performed in a quartz 96-well plate, samples were added to a reaction mixture resulting in a final concentration of 10.7 µM Tris, 2.75 M Guanidine-HCl, and 37.2 µg/mL DTNB. The absorbance at 412 nm was determined (PowerWave HT, BioTek, Winooski, VT, USA). Thiol concentration was calculated using the mM extinction coefficient for thionitrobenzoic acid (13.6).

### 2.5. Determination of Protein Concentration

The total protein content in the samples was determined by the bicinchoninic acid protein (BCA) assay (Pierce™ BCA Protein Assay Kit, ThermoFisher Scientific, Rockford, IL USA) according to manufacturer’s instructions.

### 2.6. Thioredoxin Reductase 1 (TrxR1) Expression and Activity

The relative activity of TrxR1 was determined according to the original method in which Trx-dependent reduction of insulin disulphides were measured [29]. The total value of TrxR was not determined since the purpose of the investigation was to show relative changes in TrxR activity due to medium composition. Other methods included direct reduction of DTNB and calculation of units. In the current investigation, however, the original well-established method was used in order to compare the activities in the different media incubations. In brief, 50 µg of protein from cell lysate was added to glass tubes and kept on ice. A reaction mixture was added to the tubes containing: (4-(2-hydroxyethyl)-1-piperazineethanesulfonic acid (HEPES) pH 7.45 (0.33 M), EDTA (66 mM), nicotinamide adenine dinucleotide phosphate (NADPH) (13.33 mg/mL), and insulin (3.33 mg/mL). All samples were prepared in two test tubes, where one of the two contained 10 µM of *Escherichia coli* (*E. coli*) Trx. H_2_O was added to achieve a final volume of 120 µL. The tubes were sealed and placed at 37 °C for 20 min. After the incubation the reaction was terminated by the addition of 4.84 M Guanidine-HCl containing 3.23 mg/mL DTNB. Samples were loaded onto a quartz 96-well plate, and the absorbance at 412 nm was recorded. Absorbance without Trx was subtracted from the absorbance with Trx.

### 2.7. Western Blot

50 µg of protein from cell lysate were separated on a 12% Mini-PROTEAN^®^ TGX^TM^ gel (BioRad, Hercules, CA, USA) and transferred to an Immun-Blot Polyvinylidene Difluoride (PVDF) membrane (BioRad, Hercules, CA, USA) using semi-dry transfer. Membranes were blocked in 3% BSA overnight at 4 °C, followed by incubation with TrxR1 antibody (1:1000, GeneTex, Irvine, California, USA), and β-actin (1:3000, Sigma-Aldrich, Darmstadt, Germany) for 1 h at room temperature. Membranes were washed and infrared flourescent IRDye^®^ secondary antibodies were applied (1:10,000, LI-CORLincoln, Nebraska, USA). The fluorescence was measured and analyzed using the Oddesey Fc (LI-COR^®^, Lincoln, Nebraska, USA) and the program Odyssey^®^ image studio from the same supplier. Quantification of the TrxR1 levels was determined by normalization of the fluorescence intensity to the intensity of β-actin in each sample.

### 2.8. Immunocytochemical Staining

Cells were seeded on to superfrost plus glass, placed in a rectangular multi-well dish (Nunclon, Fisher Scientific, Rockford, lL USA) and incubated for 48 h. Attached cells were washed in PBS and fixed in polyethylene-glycol (PEG) for 24 h before staining. All staining’s where performed by the accredited laboratory for Clinical Pathology and Cytology, Karolinska University Hospital (Huddinge, Stockholm, Sweden). Anti-Cytokeratin 18 (1:200 dilution) and anti-vimentin (VIM-V9, 205 mg/L, 1:2000 dilution), were both purchased from Novocastra^TM^, Leica Biosystem (Nubloch, Germany).

### 2.9. Statistical Analysis

All statistical analysis was performed using GraphPad Prism 6 (GraphPad Software Inc. La Jolla, California, USA). Normality tests and appropriate parameters were performed on the data, given the outcome, the non-parametric Kruskall–Wallis test was used to determine statistical differences between the different media.

## 3. Results

### 3.1. Cell Culture Media Influence Selenium Cytotoxicity

The toxicity and mechanisms of selenium compounds (especially selenite) have been reported in a great number of publications. However, there are large differences in reported IC_50_ values even though the experiments are performed using the same cell line (Table 1). In order to study how different media compositions influence the toxicity of selenium, cells were treated with selenite or MSC (Se-methylselenocysteine) for 48 h and the viability was determined (Figure 1). Culture and treatment of A549 cells in DMEM significantly reduced the toxicity of selenite compared to culture in RPMI or F12 (Figure 1A). In the same manner, HepG2 cells were more sensitive to selenite when cultured in MEM compared to DMEM (Figure 1C). No major differences were observed regarding the cytotoxicity of MSC for A549 cells but HepG2 cells were significantly more sensitive to MSC when cultured in F12 compared to RPMI (Figure 1D).

### 3.2. MSC Interacts with WST-1

Incubation of MSC with the WST-1 reagent resulted in a very prominent increasingly orange color. Despite clear visual cytotoxic effects as determined by light microscopy, the assay indicated continued increasing “viability” by high and increasing doses of MSC. The results clearly indicated an interaction between MSC and the WST-1 reagent thereby showing that results obtained by this method would be misinterpreted (Figure 1E,F). Since nonsense data were generated by this method, no statistical analysis was performed as the results were not valid and were incorrect.

### 3.3. Proliferation Rate and ATP Production are Dependent on Cell Culture Media Composition

To investigate how the different growth media affected the growth rate, A549 and HepG2 cells were cultured in RPMI, F12, DMEM and MEM. Cell culture in DMEM increased the proliferation rate mostly for both cell lines, and significantly in A549 cells, compared to the other media tested (Figure 2A), while culturing A549 and HepG2 cells in MEM resulted in the lowest proliferation rate. To confirm this, the ATP basal level in cells cultured during 72 h were analyzed. A549 cells cultured in MEM had significantly reduced ATP production when compared to the other cell culture media (Figure 2B).

### 3.4. Intracellular and Extracellular Thiol Content in A549 and HepG2 Cells Were Not Affected by the Different Cell Culture Media

Uptake and toxicity of three different redox active selenium compounds, selenite, selenocystine and selenodiglutathione are highly dependent on the extracellular thiol concentration and by that the cysteine recycling mediated by the glutamate/cystine (Xct) antiporter and multidrugresistant protein (MRP:s) [27].

The concentration of thiols in the studied cell culture media varied between 25 and 40 µM (Figure 3C). However, these variations did not influence the intracellular thiols of the cells (Figure 3A,B).

### 3.5. The Cell Growth Media MEM Effects the Activity and Relative Expression of Thioredoxin Reductase

Although no significant variation in the protein levels were detected, the TrxR1 activity significantly increased in A549 cells cultured in MEM compared to the other media tested (Figure 4). However, we would like to emphasize that the culture conditions were suboptimal in all media used due to the low selenium contents of the applied media and FBS. The levels were not sufficient to saturate the expression of selenoproteins. The pupose of the present study was to study any differences in TrxR expression and activity as a result of the choice of culture medium, even though saturated conditions were not obtained.

### 3.6. Selection of Cell Culture Media Changes the Expression of Markers for Epithelial and Mesenchymal Phenotype

Cells cultured for six passages in new media were cultured on glass slides, fixed and stained for epithelial (CK18) and mesenchymal (vimentin) markers. A549 cells exhibited a lower expression of CK18 when cultured in MEM and DMEM. In addition, A549 had a lower expression of Vimentin when the cells were cultured in F12 and an increased expression when cultured in DMEM (Figure 5A). These results suggested a change in phenotype in A549 cells cultured in MEM and were more pronounced in DMEM. In HepG2, the CK18 expression was slightly lower upon cultivation in RPMI, but the cells were negative for vimentin independently of the culture media (Figure 5B).

## 4. Discussion

Immortalized cells are a commonly used model system in biomedical research due to its simplicity, availability and high throughput characteristics. However, the composition of cell culture medium used varies extensively and may affect the results from biochemical, toxicological and pharmacological studies substantially depending on the experimental conditions.

In the present work we investigated the effects of four different media on the proliferation, phenotype characteristics, expression of thioredoxin reductase and cytotoxicity of selenium in two different cell lines. The results presented herein showed that cells cultured in DMEM exhibited the highest rate of proliferation while cells cultured in MEM had the lowest proliferation rate. DMEM is present in two different formulations with a high glucose level (25.0 mM) used in the present study and a low glucose formulation. The high glucose formulation is not physiological, rather it mimics the situation in diabetic patients and would thus expect to induce a number of events along with a growth drive and metabolic stress in the cells as previously shown in liver-derived cell lines [30,31]. We chose the high glucose formulation in order to study the effects of a high glucose exposure. The DMEM formulation used in this study had the highest concentration of glucose 25.0 mM as compared to MEM, RPMI and F12 with glucose concentrations of 5.56 mM, 11.1 mM and 10.01 mM, respectively. Our data demonstrated that a fold change in cell growth was reflected in the relative glucose contents of respective medium used for the four media compared in this study. High glucose has been shown to increase the proliferation rate, increase the glucose uptake and invasiveness in endometrial cancer cells, while low levels of glucose have been shown to induce cell cycle arrest and apoptosis [32]. Furthermore, the concentrations of methionine and cystine were considerably higher in DMEM compared to MEM. Methionine is important for methylation reactions where S-adenosylmethionine (SAM) is a key factor. Methylation reactions, in turn, are important regulators of cell proliferation, metabolism, biosynthesis and cell signaling. Furthermore, methionine has been shown to promote tumor cell growth while deficiency has been shown to induce cell death [3,14]. Another amino acid of importance is cysteine as it is a limiting factor in glutathione (GSH) synthesis [33]. Since GSH is an important redox buffer within the cell and many tumor cells have increased levels of GSH, the high concentration of cystine in DMEM likely favors cell proliferation due to a facilitated GSH synthesis. However, the differences demonstrated in cell proliferation may not only depend on the different levels of glucose and cysteine since the media differ greatly in other components as demonstrated by iron, for example.

Our data showed a shift in the phenotype confirmed by changes in CK18 and Vimentin expression that could indicate a gain in mesenchymal characteristics. The lung cancer cell line A549 exhibited a decreased expression of the epithelial marker CK18 when cultured in MEM and DMEM but a more pronounced expression of the mesenchymal marker Vimentin. These results were in agreement with a recent publication using a breast cancer cell line where epithelial-mesenchymal transition EMT was concluded from whole genome expression analysis after culturing cells in DMEM [26]. Another recent study showed that high glucose induced EMT was caused by increased intracellular reactive oxygen species (ROS) production, and inhibition of Trx activity, which was prevented either by treatment with the antioxidant NAC or inhibition of thioredoxin interacting protein (TXNIP) [34]. A change in phenotype due to medium composition is a serious effect that could profoundly change the outcome of any pharmacological and mechanistic studies. These observations must therefore be carefully taken into consideration when interpreting any data from cell experiments.

Thioredoxin reductase is one of 25 human selenoproteins and is embryonically lethal when not expressed [35]. TrxR1 is localized to the cytosol and important for many regulatory mechanisms and has broad substrate specificity for numerous low molecular weight compounds apart from the substrate Trx1. Among these are selenium compounds [36,37], lipid hydroperoxides [38], Q10 [39], lipoic acid [40] and ascorbate [41].

The activity of thioredoxin reductase (TrxR) was significantly higher in A549 lung cancer cells cultured in the relatively poor MEM medium containing the lowest concentration of glucose and a relatively low concentration of cystine. This composition may lead to decreased synthesis of GSH and thereby a reduced activity of GSH dependent redox enzymes that regulate the intracellular reducing environment as is often part of the malignant phenotype. On the other hand, the F12 medium contains double the amount of glucose compared to MEM, and also lipoic acid that may decrease the requirement for a high activity of TrxR.

Selenium is an essential trace element with complex biochemical and growth regulatory properties. Low levels of selenium stimulate cell growth and selenium is an essential component for the growth of cells in culture under serum free conditions. The effects of selenium are highly concentration dependent and even modest concentrations strongly inhibit the growth of cells, especially the growth of tumor cells [42]. In low concentrations selenium is considered to be an antioxidant since this element is an essential part of several peroxidases. In higher concentrations there is a shift in effects where especially redox active selenium compounds will turn into strong prooxidants causing massive ROS formation, thiol oxidation and NADPH depletion [2,42,43]. Our previous studies have shown that the uptake and the relative cytotoxicity of selenium compounds are dependent on the extracellular environment. The amount of extracellular thiols and the ability to sustain high levels of especially cysteine is one factor determining the tumor specificity of selenium compounds [27].

## 5. Conclusions

Our experiments evidently demonstrate that the cellular response to selenium is determined by the cell culture medium applied. Although a high level of extra cellular thiols, in particular cysteine, conveys an efficient uptake and a pronounced cytotoxicity of selenium, media with high basal contents of cystine did not result in any increased cytotoxicity. A549 lung cancer cells were significantly less sensitive to the redox active selenium compound selenite when cultured in DMEM. Since one prominent mechanism of selenium cytotoxicity is ROS formation due to redox cycles with thiols and oxygen [1], the composition of DMEM with high glucose and cystine levels may favor protection from selenium mediated ROS formation.

Since there is currently a focus on studies of selenium compounds as potential chemotherapeutic agents and many of these studies are performed using cell lines as model systems, our data are of the outermost importance. Thus, the selenium mediated cytotoxic effects of one cell line might be completely different depending on the medium used in the experiments. Authors should discuss the results and how they can be interpreted in light of previous studies and of the working hypotheses. The findings and their implications should be discussed in the broadest context possible. Future research directions may also be highlighted.

## Figures and Tables

**Figure 1 antioxidants-08-00130-f001:**
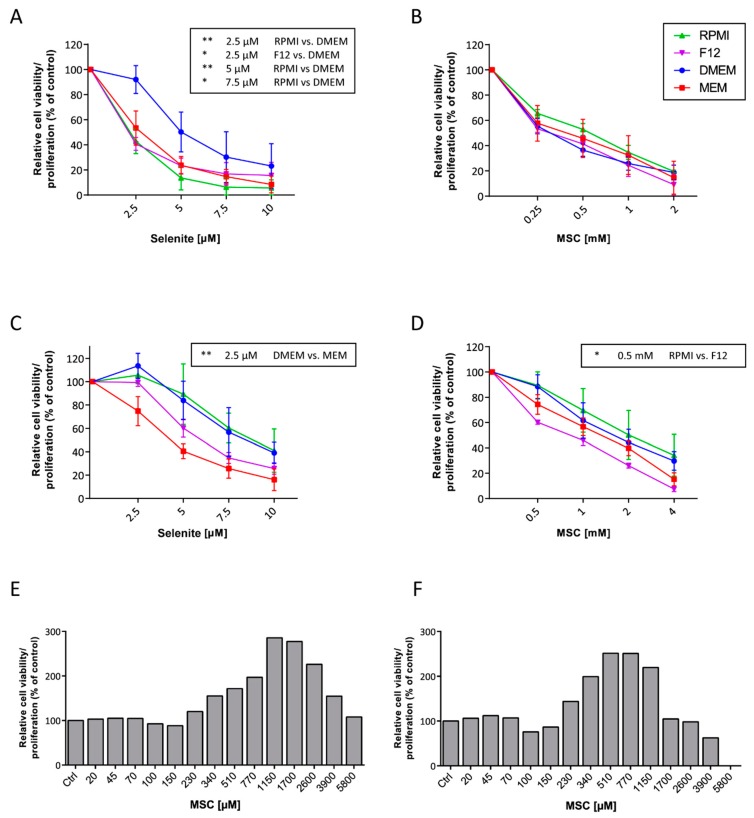
Selenite and Se-methylselenocysteine (MSC) toxicity. Effect of the different media on selenite and MSC toxicity determined by measuring ATP production and calculated as viability (%). (**A**) and (**B**) A549 cells, (**C**) and (**D**) HepG2 cells. Data is presented as mean +/− standard deviation. Statistical analysis was performed using the Kruskall–Wallis test. (* *p* < 0.05 and ** *p* < 0.01) (*n* = 3–4, all experiments were done in triplicates). Toxicity evaluation of MSC on HepG2 using WST-1 at (**E**) 24 h and (**F**) 48 h (*n* = 1).

**Figure 2 antioxidants-08-00130-f002:**
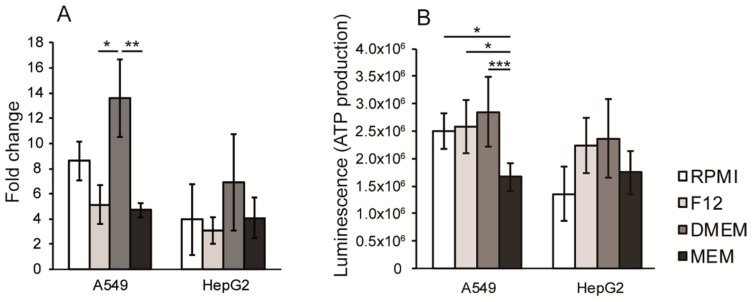
The effect on cell growth, ATP production and morphology by cell culture in different media (**A**) Cell growth after four days, shown as fold change, measured by trypan blue exclusion assay. (**B**) Base line production of ATP of cells cultured in the different media, measured at 72 h after seeding. Data is presented as mean +/− standard deviation. Statistical analysis was performed by using the Kruskall–Wallis test (* *p* < 0.05; ** *p* < 0.01; *** *p* < 0.001).

**Figure 3 antioxidants-08-00130-f003:**
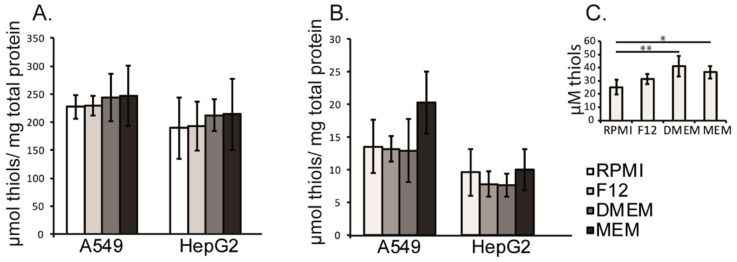
Intracellular and extracellular thiol content. The 5,5’-Dithiobis-(2-Nitrobenzoic Acid (DTNB) assay was used to measure the thiol concentration; (**A**) total intracellular thiols. (**B**) Extracellular thiols measured 5 h after the addition of fresh media. (**C**) Baseline thiol content measured in fresh media. Data is presented as mean +/− standard deviation. Statistical analysis was performed by using the Kruskall–Wallis test (*n* = 3).

**Figure 4 antioxidants-08-00130-f004:**
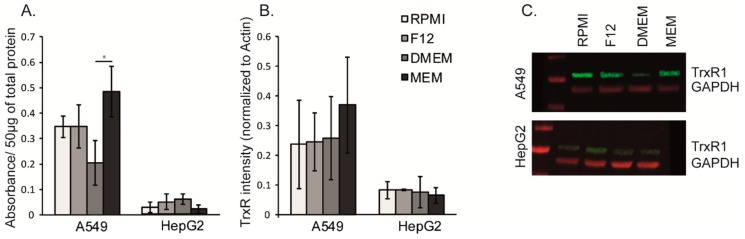
Activity and levels of TrxR. (**A**) Changes of TrxR activity influenced by cell culture media determined with TrxR-assay and (**B**) TrxR1 protein concentration analyzed by western blot. Quantification was made with three independent experiments. (**C**) Representative Western Blot image. Data is presented as mean +/− standard deviation. Statistical analysis was performed by using the Kruskall-Wallis test (* *p* < 0.05).

**Figure 5 antioxidants-08-00130-f005:**
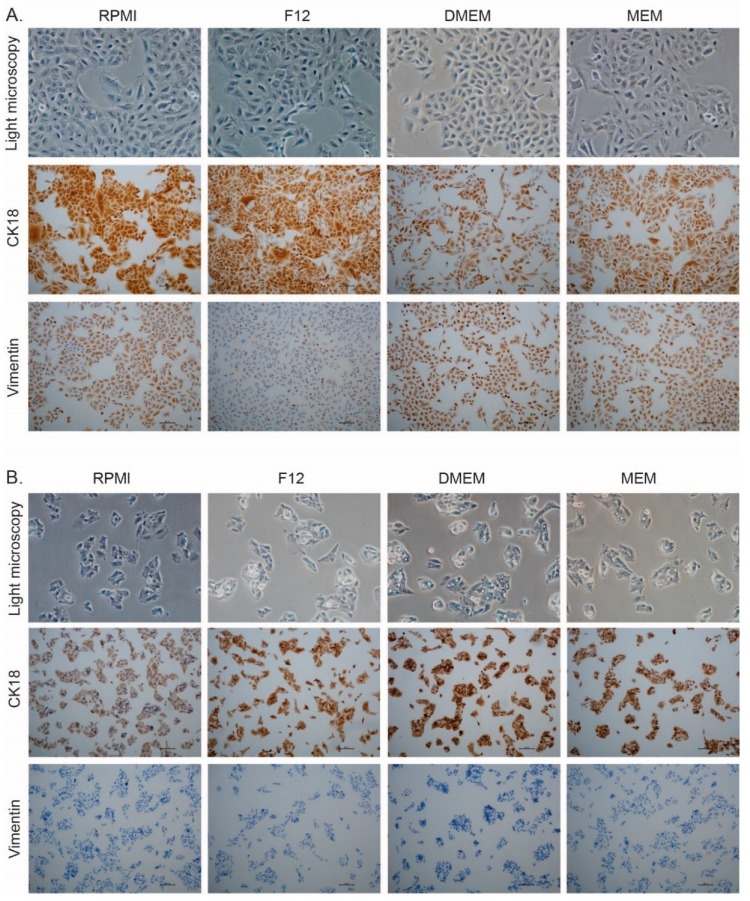
Morphology and immunocytochemical staining of cells cultivated in the different media. (**A**) Light microscopy (40×), CK18 and Vimentin staining of A549 cells. (**B**) Light microscopy (40×), CK18, and Vimentin staining of HepG2 cells.

**Table 1 antioxidants-08-00130-t001:** Varying IC_50_ values after 24 h of treatment with selenite or seleno-methylselenocysteine (MSC) shown in different studies.

**Selenite**	**Medium**	**FBS**	**Antibiotics**	**IC_50_**	**Evaluated by**	**Ref.**
**Cell Line**						
A549	RPMI 1640	10%	Yes	~6 µM	MTT	[3]
A549	DMEM	10%	No	8.2 µM	Sulforhodamine	[4]
A549	DMEM+ non ess. AA	2%	Yes	5 µM = 85% viability	MTT	[5]
HepG2	1:1 DMEM:F12	10%	Yes	25.7 µM	MTT	[6]
HepG2	DMEM	10%	Yes	~10 µM	MTT (48 h)	[7]
HepG2	MEM	10%	Yes	~7.5 µM	Tunnel-assay	[8]
Huh7	DMEM	10%	Yes	~20 µM	XTT	[9]
HL-60	RPMI 1640	10%	Yes	20 µM	DNA fragmentation	[10]
NB 4	RPMI 1640	10%	Yes	20 µM	Trypan blue	[11]
**MSC**	**Medium**	**FBS**	**Antibiotics**	**IC_50_**	**Evaluated by**	**Ref.**
**Cell Line**						
HT 29/ SW480/ SW620	DMEM	10%	Yes	~64/~32/ ~90µM	MTT (48 h)	[12]
HL-60	RPMI 1640	10%	Yes	50 µM	DNA fragmentation	[10]

RPMI 1640: Roswell Park Memorial Institute 1640 Medium; DMEM: Dulbecco’s Modified Eagle’s Medium; non ess: non-essential; AA: amino acids; F12: Ham’s F-12 Nutrient Mixture; MEM: Minimum Essential Medium; FBS: Fetal bovine serum; **IC_50_**: half-maximal inhibitor concentration; MTT: 3-(4,5-dimethylthiazol-2-yl)-2,5-diphenyltetrazolium bromide assay; XTT: (2,3-bis-(2-methoxy-4-nitro-5-sulfophenyl)-2H-tetrazolium-5-carboxanilide) assay.

**Table 2 antioxidants-08-00130-t002:** Variation in media selection for the same cell line.

Cell Line	Recommended Medium by Suppliers	RPMI	F12	DMEM	MEM	DMEM/F12
**A549**	DMEM or F12	[3,13]	[14]	[4,5]	[15,16]	
**HepG2**	MEM	[17,18,19]		[15,20,21]	[22]	[23]

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
