# Peer review of "The Cell Culture Medium Affects Growth, Phenotype Expression and the Response to Selenium Cytotoxicity in A549 and HepG2 Cells"

_antioxidants, 2019, doi:10.3390/antiox8050130_

Round 1

Reviewer 1 Report

Minor Poinst:

"medium" is singular; "media" is plural.  Therefore, for example, on line 99, "..fresh media was added.." should be changed to "fresh media were added" or "fresh medium was added".

Along the same line, "data" should be treated as a plural (see for example line 143).

Author Response

Reviewer 1

Comments and Suggestions for Authors

Minor Points:

"medium" is singular; "media" is plural.  Therefore, for example, on line 99, "..fresh media was added.." should be changed to "fresh media were added" or "fresh medium was added".

Changes have been done accordingly throughout the manuscript.

Along the same line, "data" should be treated as a plural (see for example line 143

Changes have been done accordingly throughout the manuscript.

Reviewer 2 Report

Selenius et al. investigated the effect of culture media on diffferent parameters (growth, cytotoxicity of Se compounds, TrxR expression/activity, morphology) for two cell lines derived from lung and liver carcinoma. They report that the culture medium has a profound and partly cell-specific influence on some of the selected parameters, and thus, the cell culture medium should be taken into account as one important factor when judging and comparing results from in vitro studies.In general, I found the manuscript interesting and well written. The following questions/suggestions are meant to further improve its quality:

1) The title of the paper is somewhat mistakable as it refers only to the first part of the study. From the title, I expected to see more experiments on the influence of culture media on the response of the cells to selenium. Thus, I suggest to broaden the title, e.g. " The culture medium affects growth, morphology and response to redox-active selenium compounds of A549 and HepG2 cells"

2) Lines 136/137: The authors state that "HepG2 cells were more sensitive to selenite when cultured in DMEM compared to MEM (Figure 1C)". To my mind, it looks vice versa in Fig. 1C; selenite was more cytotoxic in MEM compared to DMEM. Please check! Also, please replace "Se" in Figs. 1A and 1C by "selenite" (Se means usually selenium).

3) Line 168: The authors state that "Selenium uptake and toxicity is highly dependent on the extracellular thiol concentrations". Please give a literature reference to support this statement. To my mind, it is too general, as cellular uptake of different Se compounds is through different transporters, and different Se compounds have very different cytotoxicity.

4) Expression/activity of TrxR: Please also describe in Material and methods how the immunoblotting for TrxR1 detection was performed. Please provide a representative TrxR1 immunoblot in addition to the graph in Fig. 4B. For Fig. 4A, please calculate and show the specific TrxR activity (given in mU/mg protein), as it is usually done in the literature. To my mind, it would be nice to include data here on TrxR expression/activity in cells supplemented with Se in the four different culture media. The experimental design of the authors is here suboptimal, as culture media and FBS do not contain enough Se for optimal biosynthesis of selenoproteins (please see M. Leist et al. FRBM 1996)

Lines 245/246: Please make it more clear to the reader that Se is per se not an antioxidant at all. Cytoprotective actiions of Se are mediated by its incorporation (in form of Sec) into antioxidant selenoenzymes such as glutathione peroxidases.

Lines 202-210 (Higher proliferation of the cells in DMEM as compared to MEM and the potential effect of glucose): Here, it should be stated in addition that DMEM is available in two formulations, DMEM high glucose and DMEM low glucose. In fact, HepG2 cells are usually grown in DMEM low glucose, as 25 mM glucose is not physiological (such high glucose concentrations are rather found in the blood of non-treated diabetes patients) and it is already sufficient to induce intracellular triglycerid accumulation in the HepG2 cells (which is in vivo characteristic for the pathogenesis of a fatty liver) (please see NA Hoang et al. Eur J Nutr 2018).

Author Response

Reviewer 2

Comments and Suggestions for Authors

Selenius et al. investigated the effect of culture media on diffferent parameters (growth, cytotoxicity of Se compounds, TrxR expression/activity, morphology) for two cell lines derived from lung and liver carcinoma. They report that the culture medium has a profound and partly cell-specific influence on some of the selected parameters, and thus, the cell culture medium should be taken into account as one important factor when judging and comparing results from in vitro studies.In general, I found the manuscript interesting and well written. 

The following questions/suggestions are meant to further improve its quality:

1) The title of the paper is somewhat mistakable as it refers only to the first part of the study. From the title, I expected to see more experiments on the influence of culture media on the response of the cells to selenium. Thus, I suggest to broaden the title, e.g. " The culture medium affects growth, morphology

The title has been changed to The cell culture medium affects growth, morphology and the response to selenium cytotoxicity in A549 and HepG2 cells”

2) Lines 136/137: The authors state that "HepG2 cells were more sensitive to selenite when cultured in DMEM compared to MEM (Figure 1C)". To my mind, it looks vice versa in Fig. 1C; selenite was more cytotoxic in MEM compared to DMEM. Please check! Also, please replace "Se" in Figs. 1A and 1C by "selenite" (Se means usually selenium).

We are grateful to this valuable remark and have made changes accordingly.

3) Line 168: The authors state that "Selenium uptake and toxicity is highly dependent on the extracellular thiol concentrations". Please give a literature reference to support this statement. To my mind, it is too general, as cellular uptake of different Se compounds is through different transporters, and different Se compounds have very different cytotoxicity.

The referee is correct that this statement is too general. In fact, we in a previous publication (Olm et al., 2009) showed that the extracellular environment had a major impact on the uptake of three different redox-active selenium compounds, namely, selenite, selenocystine and selenodiglutathione. We have clarified this in the revised version of the manuscript.

4) Expression/activity of TrxR: Please also describe in Material and methods how the immunoblotting for TrxR1 detection was performed. Please provide a representative TrxR1 immunoblot in addition to the graph in Fig. 4B. For Fig. 4A, please calculate and show the specific TrxR activity (given in mU/mg protein), as it is usually done in the literature. To my mind, it would be nice to include data here on TrxR expression/activity in cells supplemented with Se in the four different culture media. The experimental design of the authors is here suboptimal, as culture media and FBS do not contain enough Se for optimal biosynthesis of selenoproteins (please see M. Leist et al. FRBM 1996)

The immunoblotting for TrxR has been added in the MM section and an immunoblot has been added denoted Fig 4C. The activity of TrxR was determined as a relative activity using the classical method including Trx dependent reduction of insulin disulphides. The method (Holmgren and Björnstedt, 1995, Methods Enz, Biothiols) is widely used to assess the relative activity and units are generally not provided as compared to other methods employing the reduction of DTNB where the units are defined as 1 micromol of TNB formed per minute. In the classical original method, generally activities are compared according to the results shown in the present paper and if the total amount is required and of relevance to the results, the absorbances are compared to a standard curve and the amounts calculated as ng/mg of protein. Since the total level of TrxR as absolute values are not relevant in the present comparison a standard curve was not provided. The method has been more thoroughly described in the revised version.

The referee raises a very valid point concerning suboptimal conditions for the activity of TrxR. It is perfectly true that the amount of selenium in the different media and FCS is not sufficient for saturated conditions. However, the point of the study was to investigate the activity of TrxR after cultivation of cells in the different media, and thus not to compare conditions of selenium excess. The experimental conditions are justified in the revised version in the results and discussion sections.

Lines 245/246: Please make it more clear to the reader that Se is per se not an antioxidant at all. Cytoprotective actiions of Se are mediated by its incorporation (in form of Sec) into antioxidant selenoenzymes such as glutathione peroxidases.

We have clarified this in the manuscript.

Lines 202-210 (Higher proliferation of the cells in DMEM as compared to MEM and the potential effect of glucose): Here, it should be stated in addition that DMEM is available in two formulations, DMEM high glucose and DMEM low glucose. In fact, HepG2 cells are usually grown in DMEM low glucose, as 25 mM glucose is not physiological (such high glucose concentrations are rather found in the blood of non-treated diabetes patients) and it is already sufficient to induce intracellular triglycerid accumulation in the HepG2 cells (which is in vivo characteristic for the pathogenesis of a fatty liver) (please see NA Hoang et al. Eur J Nutr 2018).

This is a very valid remark and a couple of sentences have been added along with two more references.

Reviewer 3 Report

This paper examines the cytotoxic effects of selenium compounds and suggest that this might be related to the constituents of culture media. Two cell types were cultured in four different media media. The results are intriguing, but I feel further investigation is warranted before publication.

1)      Trypan blue exclusion is a measure of cell viability, not proliferation as stated in Figure 2. How were growth rates determined?

2)      Were the cells of equal confluency when used. We find that the degree of confluency is critical in determining response to oxidative insult. This should be kept below 80%.

3)      The authors suggest that the effect is due to high glucose in DMEM and high cysteine.  A critical experiment is to take the low glucose and low cysteine MEM and add additional glucose and cysteine to see if this explains the biological effect.

4)      Was the MSC and selenite treatment conducted in complete media with 10%FBS?

5)      The authors suggest that the negative effect is due to oxidative stress and generation of ROS. In this case the experiments should be repeated at lower oxygen tensions, perhaps 5% or 8%.  Cell culture at 20% O2 is not physiological.

6)      There is no mention of how many times these experiments were done. The study needs to quote n numbers and whether data points were determined in duplicate or triplicate. How was the data analysed? For example Figure 1E , it is not satisfactory to have 1 representative experiment.

Author Response

Reviewer 3

Comments and Suggestions for Authors

This paper examines the cytotoxic effects of selenium compounds and suggest that this might be related to the constituents of culture media. Two cell types were cultured in four different media media. The results are intriguing, but I feel further investigation is warranted before publication.

1)      Trypan blue exclusion is a measure of cell viability, not proliferation as stated in Figure 2. How were growth rates determined?

This has now been clarified in the materials and methods section. The fold change in growth rate relates to the number of cells seeded at day 0.

2)      Were the cells of equal confluency when used. We find that the degree of confluency is critical in determining response to oxidative insult. This should be kept below 80%

Yes they were, the seeding densities chosen were optimised before conducting the experiments.

3)      The authors suggest that the effect is due to high glucose in DMEM and high cysteine.  A critical experiment is to take the low glucose and low cysteine MEM and add additional glucose and cysteine to see if this explains the biological effect.

DMEM high glucose was used in these experiments which is not a physiological glucose level but rather mimics the level in diabetic patients. DMEM is a considerable richer medium as compared to MEM and several important nutrients are lacking in MEM as compared to DMEM, not just glucose and cystine. We have in the revised version clarified the use of high glucose DMEM and provided references to that this glucose level is not physiological and may affect the growth and reaction to xenobiotics of especially liver cells. The comparisons were made between DMEM, MEM, RPMI and F12, of which DMEM is exceptional in terms of glucose 25.0 mM as compared to 5.56 mM, 11.1 mM and 10.01 mM respectively. In fig 2A the fold change in cell growth is reflected in the relative glucose contents of respective medium used. We do agree that the conclusions concerning the glucose level are to extensive and therefore we have in the revised version changed the section concerning the glucose data and added new references concerning the effects of glucose.

4)      Was the MSC and selenite treatment conducted in complete media with 10%FBS?

Yes.

5)      The authors suggest that the negative effect is due to oxidative stress and generation of ROS. In this case the experiments should be repeated at lower oxygen tensions, perhaps 5% or 8%.  Cell culture at 20% O2 is not physiological.

The experiments were performed at normoxic conditions which is a physiologic level. Several publications indicate that a leading mechanism of selenium cytotoxicity is ROS-formation and considerable oxidative stress. This has been clarified in the revised version and references are added.

6)      There is no mention of how many times these experiments were done. The study needs to quote n numbers and whether data points were determined in duplicate or triplicate. How was the data analysed? For example Figure 1E , it is not satisfactory to have 1 representative experiment.

The n-number has been provided in the revised version. Concerning the WST-1 data, these data are just provided in order to demonstrate that MSC is interfering with the reagent to prevent other scientist to use a non-feasible detection method of cell viability. Since the results using this method is unpredictable and of a nonsense character statistics are impossible to perform. For clarity this has been explained in the revised version and an additional experiment is added denoted figure 1 F.  

Reviewer 4 Report

This paper writing is an evaluation of conditions of cell culture media affect the toxic effect of selenite or MSC on human lung carcinoma A549 and Hepatoma HepG2. This paper may raise interest among readers and encourage further studies.

Additional comments are presented below:

authors should provide the levels of elemental Se in the present study.

it is suggested that "selenite" instead of initialism "Se"  

the statistical analysis must be documented.

Author Response

Reviewer 4

Comments and Suggestions for Authors

This paper writing is an evaluation of conditions of cell culture media affect the toxic effect of selenite or MSC on human lung carcinoma A549 and Hepatoma HepG2. This paper may raise interest among readers and encourage further studies.

Additional comments are presented below:

1. authors should provide the levels of elemental Se in the present study.

Often mass units are used in toxicity assessments of selenium and in these situations comparisons to mass units of elemental is appropriate and necessary for comparisons. Since in the present study all values are provided using the molar unit we disagree with this referee that the level of elemental selenium should be provided. The molar units for selenite and MSC are identical to the molar units of elemental selenium.

2. it is suggested that "selenite" instead of initialism "Se"  

The manuscript has been revised accordingly and ”Se” is replaced by selenite

3. the statistical analysis must be documented.

A section describing the statistical analysis has been added under materials and methods. Thank you for highlighting this.

Round 2

Reviewer 2 Report

Thank you for your reply. Everything is well done. I do not have any other questions or concerns.

Reviewer 3 Report

I am happy to accept this modified version of the paper. I still believe that the experiments should be repeated at 5% O2 which is more physiological. Cell culture at 5%CO2 contains approximately 20% O2 which is NOT normoxic. Cell culture at 20% O2 with 25mM Glucose will generate oxidative stress and cells adapt to this after long term culture in DMEM. Changing media, glucose conc or O2 levels always causes cell stress. .

Perhaps these points could be explored in a subsequent study.